# Severity of Atelectasis during Bronchoscopy: Descriptions of a New Grading System (Atelectasis Severity Scoring System—“ASSESS”) and At-Risk-Lung Zones

**DOI:** 10.3390/diagnostics14020197

**Published:** 2024-01-16

**Authors:** Asad Khan, Sami Bashour, Bruce Sabath, Julie Lin, Mona Sarkiss, Juhee Song, Ala-Eddin S. Sagar, Archan Shah, Roberto F. Casal

**Affiliations:** 1Department of Pulmonary Medicine, The University of Texas MD Anderson Cancer Center, Houston, TX 77030, USAbsabath@mdanderson.org (B.S.); jlin5@mdanderson.org (J.L.); 2Department of Anesthesia and Peri-Operative Medicine, The University of Texas MD Anderson Cancer Center, Houston, TX 77030, USA; msarkiss@mdanderson.org; 3Department of Biostatistics, The University of Texas MD Anderson Cancer Center, Houston, TX 77030, USA; 4Department of Internal Medicine, King Faisal Specialist Hospital and Research Center, Madinah 42523, Saudi Arabia; alaeddin.sagar@gmail.com; 5Department of Onco-Medicine, Banner MD Anderson Cancer Center, Gilbert, AZ 85234, USA; archan.shah@bannerhealth.com

**Keywords:** atelectasis, bronchoscopy, general anesthesia

## Abstract

Atelectasis during bronchoscopy under general anesthesia is very common and can have a detrimental effect on navigational and diagnostic outcomes. While the intraprocedural incidence and anatomic location have been previously described, the severity of atelectasis has not. We reviewed chest CT images of patients who developed atelectasis in the VESPA trial (Ventilatory Strategy to Prevent Atelectasis). By drawing boundaries at the posterior chest wall (A), the anterior aspect of the vertebral body (C), and mid-way between these two lines (B), we delineated at-risk lung zones 1, 2, and 3 (from posterior to anterior). An Atelectasis Severity Score System (“ASSESS”) was created, classifying atelectasis as “mild” (zone 1), “moderate” (zones 1–2), and “severe” (zones 1–2–3). A total of 43 patients who developed atelectasis were included in this study. A total of 32 patients were in the control arm, and 11 were in the VESPA arm; 20 patients (47%) had mild atelectasis, 20 (47%) had moderate atelectasis, and 3 (6%) had severe atelectasis. A higher BMI was associated with increased odds (1.5 per 1 unit change; 95% CI, 1.10–2.04) (*p* = 0.0098), and VESPA was associated with decreased odds (0.05; 95% CI, 0.01–0.47) (*p* = 0.0080) of developing moderate to severe atelectasis. ASSESS is a simple method used to categorize intra-bronchoscopy atelectasis, which allows for a qualitative description of this phenomenon to be developed. In the VESPA trial, a higher BMI was not only associated with increased incidence but also increased severity of atelectasis, while VESPA had the opposite effect. Preventive strategies should be strongly considered in patients with risk factors for atelectasis who have lesions located in zones 1 and 2, but not in zone 3.

## 1. Introduction

Atelectasis during general anesthesia has been a well-studied phenomenon in the surgical literature, described many decades ago [1,2,3,4,5,6]. But it was not until recently, with the advent of cone beam computed tomography (CBCT) guidance for peripheral bronchoscopy, that atelectasis was identified as a common phenomenon during bronchoscopy under general anesthesia. After the initial report by Casal et al. during a pilot study on CBCT-guided bronchoscopy, its incidence was further studied by the same group of scientists in the I-LOCATE trial [7,8]. The latter enrolled patients undergoing bronchoscopy under general anesthesia (GA) and evaluated posteriorly located bronchial segments with radial probe endobronchial ultrasound (RP-EBUS), demonstrating that greater than fifty percent of the patients developed atelectasis after 30 min of general anesthesia in the lower lobes [8]. I-LOCATE provided a detailed anatomical map with the incidence of atelectasis in each bronchial segment that was at risk. A higher body mass index (BMI) and longer time under general anesthesia were, not surprisingly, associated with more atelectatic bronchial segments.

Unlike the surgical population, where atelectasis is associated with perioperative complications, in patients undergoing peripheral bronchoscopy, the deleterious effect of atelectasis is mainly intra-operative. Atelectasis can result in false positive RP-EBUS images, obscure the target lesion on intraprocedural CBCT images, and increase CT-to-body divergence, thereby affecting both the navigation and diagnostic yield of bronchoscopy [9,10,11,12]. A multicenter randomized controlled trial of standard fluoroscopy-guided bronchoscopy versus a thin bronchoscope with RP-EBUS conducted by Tanner and coworkers reported one of the greatest gaps between navigation (97%) and diagnostic yield (50%), which could be partly explained by atelectasis and false-positive images [13]. Thus, ventilatory and positional strategies have been proposed in order to prevent this unwanted phenomenon [14,15,16,17,18,19,20]. The VESPA trial (Ventilatory Strategy to Prevent Atelectasis) was a multicenter randomized controlled trial which compared standard ventilation through a laryngeal mask airway (LMA) with a 100% fraction of inspired oxygen (FiO_2_) and a positive end-expiratory pressure (PEEP) of 0 cm H2O vs. VESPA, consisting of endotracheal intubation followed by a recruitment maneuver, FiO_2_ titration (<100%), and a PEEP of 8–10 cm H_2_O [14]. All patients underwent a chest CT of the lung bases and a survey for atelectasis after artificial airway insertion (Time 1), followed by a second survey 20–30 min later (Time 2). Atelectasis was deemed present when the CT scans revealed an area of densely consolidated lung parenchyma of at least 2 cm (measured as a straight line perpendicular to the chest wall from the outermost to the innermost edge of the consolidated area). Seventy-six patients were analyzed, with thirty-eight in each group. The proportion of patients with any atelectasis (unilateral or bilateral) detected through a chest CT at Time 2 was 84.2% (95% CI, 72.6–95.8%) in the control group and 28.9% (95% CI, 15.4–45.9%) in the VESPA group (*p* < 0.0001). The proportion of patients with bilateral atelectasis at Time 2 was 71.1% (95% CI, 56.6–85.5%) in the control group and 7.9% (95% CI, 1.7–21.4%) in the VESPA group (*p* < 0.0001). No differences were found in the rate of complications. While the VESPA trial confirmed with CBCT images the worrisome high rate of intraprocedural atelectasis that was previously described in the I-LOCATE trial with RP-EBUS, the simplified definition of atelectasis utilized for the VESPA trial did not allow us to assess the severity of atelectasis and the regions of the lungs that are at the highest risk. In the current study, we specifically reviewed all chest CT images of the patients who were considered to have atelectasis in the VESPA trial to categorize the severity of atelectasis and describe the lung zones that were more commonly affected.

## 2. Materials and Methods

This study was performed at the University of Texas MD Anderson Cancer Center with Institutional Review Board (2019-0387) approval. Records and CT images from patients who were deemed to have atelectasis in the VESPA trial (NCT04311723) conducted between July 2020 and December 2021 were assessed. Baseline patient and procedure characteristics were extracted. CT scans obtained both at Time 1 (immediately after airway insertion) and Time 2 (20–30 min after Time 1) were reviewed.

The main goals of this study were to describe the severity of atelectasis encountered in the VESPA trial and to delineate at-risk zones. In order to do so, three parallel lines (A, B, and C) that divided each base into three zones (1, 2, and 3) were drawn (Figure 1). Line A was drawn horizontally at the level of the parietal pleura (at its most posterior point), and Line C was drawn horizontally at the most ventral edge of the vertebral body. The distance between these two lines was measured and divided by 2 to then draw Line B mid-way between the other two lines (and parallel to them). The lung zone between Lines A and B was called zone 1, the one between Lines B and C was called zone 2, and the lung area anterior to Line C was called zone 3. An Atelectasis Severity Scoring System (“ASSESS”) was created, classifying atelectasis as ASSESS I or “mild” (zone 1), ASSESS II or “moderate” (zones 1–2), and ASSESS III or “severe” (zones 1–2–3) (Figure 1 and Figure 2). Chest CT images were reviewed (scrolling through all CT images for each lung base), and the highest score for each lung base was recorded. Each lung zone was considered involved when we found an area of dense consolidation of at least 1 cm over the corresponding line. When patients had bilateral atelectasis, the highest score was recorded for the per patient analysis.

Patient demographic characteristics are summarized using descriptive statistics (mean ± SD or median and interquartile range—IQR—for continuous variables and frequency for categorical variables). To evaluate the factors associated with developing moderate to severe atelectasis, we used univariate and multivariable logistic regression models. The Hosmer–Lemeshow test was used to evaluate the goodness of fit of the model. A *p*-value of less than 0.05 indicated statistical significance. SAS version 9.4 (SAS Institute Inc., Cary, NC, USA) was used for data analysis.

## 3. Results

A total of 43 patients who developed atelectasis in the VESPA trial were included in this study. A total of 32 patients were in the control arm, and 11 were in the VESPA arm. The mean age was 64.75 years (SD 11.5 years), 24 patients were male (56%), and the mean BMI was 30.37 kg/m^2^ (SD 3.73). The median time from anesthesia induction to intubation was 4 min (IQR 3 to 7 min). The time from anesthesia induction to the atelectasis survey (second survey, Time 2 in VESPA trial) was 43.1 min (SD 4.9 min). The per patient analysis showed that 20 patients had mild atelectasis (47%), 20 had moderate atelectasis (47%), and 3 had severe atelectasis (6%). Zone 1 was affected in all patients, zone 2 was affected in 23 patients (53%), and zone 3 was affected in 3 patients (6%). A total of 74 CT images of atelectatic lung bases (31 patients had bilateral and 12 had unilateral atelectasis) were analyzed. In the per atelectatic lung base analysis, atelectasis was mild in 41 bases (55%), moderate in 29 bases (39%), and severe in 4 lung bases (6%). The univariate and multivariate logistic regression analyses are depicted in Table 1 and Table 2. Based on the multivariable logistic regression model, a higher BMI was associated with increased odds (1.5 per 1 unit change; 95% CI, 1.10–2.04) (*p* = 0.0098) of developing moderate to severe atelectasis. On the other hand, VESPA was protective, with decreased odds (0.05; 95% CI, 0.01–0.47) (*p* = 0.0080) of developing moderate to severe atelectasis. There was no association between age (0.98 per 1 unit change; 95% CI, 0.93–1.03) (*p* = 0.3956), gender (female 1.00, male 1.06; 95% CI, 0.32–3.56) (*p* = 0.9202), time from induction to intubation age (0.85 per 1 unit change; 95% CI, 0.66–1.10) (*p* = 0.2205), or time spent under general anesthesia (0.89 per 1 unit change; 95% CI, 0.78–1.02) (*p* = 0.1054) and developing moderate to severe atelectasis.

## 4. Discussion

The development of atelectasis during bronchoscopy under general anesthesia is a common and unwanted phenomenon that can negatively influence our ability to navigate and perform biopsies on peripheral lung lesions. While its incidence has been well studied, to the best of our knowledge, a qualitative assessment has not been performed. This is the first study to propose a scoring system (ASSESS) to delineate lung zones that are at risk of atelectasis and to describe the severity of atelectasis during bronchoscopy under general anesthesia. Our findings allow us to better understand this phenomenon and identify the lung zones that are at high risk, helping us recognize when a strategy to prevent atelectasis is indicated. ASSESS may provide a more accurate tool and a common language for future comparative studies of different strategies to prevent atelectasis.

A few studies describing strategies to prevent atelectasis have been published to date [14,17,18]. While these studies reported a substantial decrease in the incidence of intra-bronchoscopy atelectasis with the respective ventilatory strategies, the severity of atelectasis was not reported in any of these studies. In our current study, we demonstrated that when atelectasis occurs (with or without a ventilatory strategy to prevent it), it is mild in 47% of the patients, moderate in another 47%, and severe in only 6% of patients. We also demonstrated that a high BMI is not only associated with a higher incidence of atelectasis but higher severity as well. Importantly, VESPA not only reduced the incidence of atelectasis but also its severity. VESPA is now the only preventive strategy to have been demonstrated to do so. These data, along with the delineation of at-risk zones 1, 2, and 3, can help a proceduralist identify patients who will benefit the most from a strategy to prevent atelectasis. For example, most patients with lung nodules located in zone 1 will benefit from a strategy to prevent atelectasis, since this zone will be affected 100% of the time if atelectasis develops. Patients with lung nodules in zone 2 may only benefit from a preventive strategy if they also have high BMIs predisposing them to develop atelectasis. On the other hand, patients with lung nodules in zone 3 may not benefit from a strategy to prevent atelectasis, since this area of the lung is rarely affected, even when atelectasis occurs. Of course, these are assumptions based on the scant available data described above and common sense. A prediction model should be developed and validated in order to make any recommendations, and we hope the data from our studies will allow us to construct one and provide accurate guidance to bronchoscopists in the future.

While the I-LOCATE trial provided us with a detailed anatomical map with the incidence of atelectasis in each bronchial segment, it is not always easy to delineate these segments in chest CT images [8]. Our simplified method for the delineation of at-risk lung zones utilized for the grading of atelectasis in ASSESS requires less anatomical interpretation, no additional technology, and only takes a few seconds to implement. We hope this provides an easy-to-use tool for less experienced bronchoscopists.

Establishing a common language to more accurately describe this development of atelectasis during bronchoscopy under general anesthesia is of paramount importance. Our methodology can be easily applied in comparative studies of strategies to prevent atelectasis, not only between treatment arms of the same study but also to compare results of various strategies evaluated in different trials. We are currently conducting a randomized controlled study of VESPA versus the Lateral Decubitus Strategy (LADS) to prevent atelectasis in patients undergoing robotic bronchoscopy for lung nodules located in zones 1 and 2 (NCT05714033). While the primary outcome is atelectasis obscuring targets, ASSESS will be utilized to compare the severity of atelectasis between the two groups.

As with any study, the current one also has limitations. Among the main limitations, we need to recognize that the CBCT images from the VESPA trial were mostly obtained with a mobile CBCT that was only able to scan the lower half of each hemithorax. The upper lobes were not evaluated in most of the cases, and the severity of atelectasis in those areas could not be described. While the same lung zones can be delineated in the upper lobes, how often each lung zone is involved when atelectasis occurs in the upper lobes is not known. Fortunately, atelectasis in the upper lobes is much less frequent. The I-LOCATE trial found the incidence rates of atelectasis in the posterior segments of the right upper lobe and left upper lobe to be 19.3% and 16.4%, respectively (without utilizing any strategy to prevent atelectasis) [8]. Another limitation of our study is the sole inclusion of densely consolidated atelectasis, leaving ground-glass opacities (GGOs) outside of the scoring system. The GGOs were not included because the relatively poor quality of the mobile CBCT images, particularly in patients with high BMIs, precluded an accurate analysis. Nevertheless, densely consolidated atelectasis is of greater importance since this is the type that will more likely obscure our targets, which are mostly solid or semisolid. For future studies, if CBCT imaging with a higher image quality is being used and if GGOs want to be described, we suggest utilizing the same ASSESS grading, adding a lowercase “g” for GGO or “s” for solid atelectasis (i.e., ASSESS g2, meaning atelectasis in the form of GGO reaching zone 2). Our scoring system also leaves out small segmental or subsegmental atelectasis that occurs from wedging the bronchoscope, but the incidence rate of this is low in peripheral bronchoscopy due to the very small diameter of newer bronchoscopes and robotic catheters. 

In summary, ASSESS is a simple method used to categorize intra-bronchoscopy atelectasis, which allows for a qualitative description of this phenomenon. ASSESS may provide a more accurate tool and a common language for future comparative studies of different strategies to prevent atelectasis. In the VESPA trial, a higher BMI was not only associated with an increased incidence but also increased severity of atelectasis, while VESPA had the opposite effect. Based on our findings, strategies to prevent atelectasis should be strongly considered in patients with risk factors for atelectasis who have lung lesions located in zones 1 and 2, but not in zone 3.

## Figures and Tables

**Figure 1 diagnostics-14-00197-f001:**
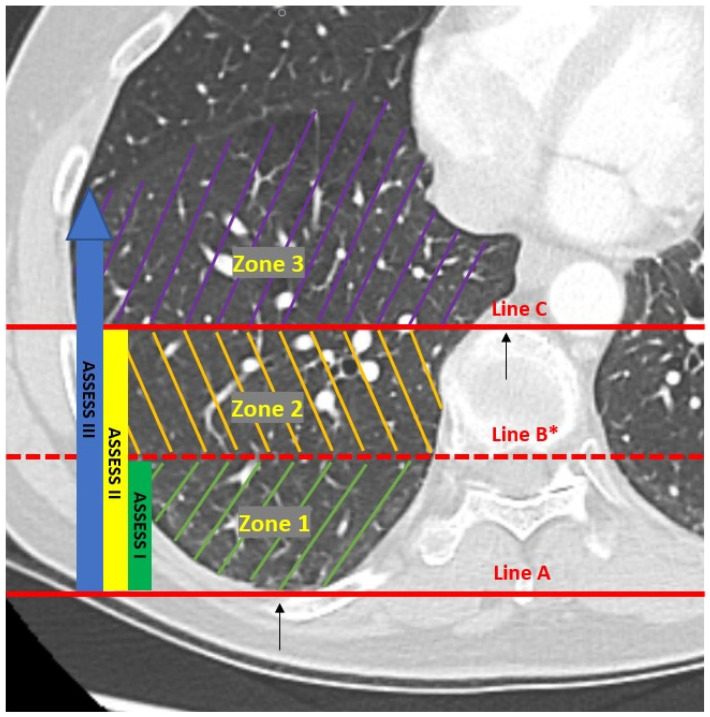
Atelectasis Severity Scoring System (ASSESS). * Line B = distance between Lines A and C/2.

**Figure 2 diagnostics-14-00197-f002:**
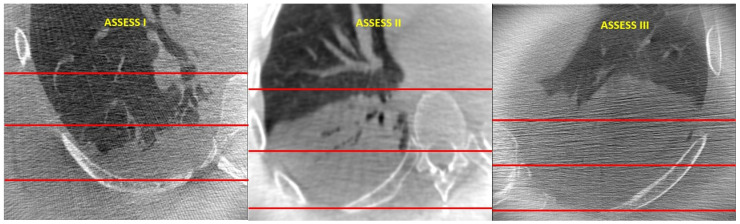
Chest CT images demonstrating atelectasis of different degrees. ASSESS I = “mild”; ASSESS II = “moderate”; ASSESS III = “severe”.

**Table 1 diagnostics-14-00197-t001:** Univariate logistic regression analysis in predicting moderate to severe atelectasis.

Covariate	Level	OR (95% CI)	*p*-Value
Age	In 1 Unit Change	0.98 (0.93–1.03)	0.3956
BMI	In 1 Unit Change	1.22 (1.00–1.48)	0.0527
Time from Induction to Intubation	In 1 Unit Change	0.85 (0.66–1.10)	0.2205
Time from Induction to Time 2 Atelectasis Survey	In 1 Unit Change	0.89 (0.78–1.02)	0.1054
Allocation Group	Control	1.00	
	VESPA	0.23 (0.05–1.02)	0.0525
Gender	Female	1.00	
	Male	1.06 (0.32–3.56)	0.9202
Time 1 Atelectasis Survey Score	0	1.00	
	1–2	2.14 (0.61–7.53)	0.2364

**Table 2 diagnostics-14-00197-t002:** Multivariate logistic regression analysis in predicting moderate to severe atelectasis.

Covariate	Level	OR (95% CI)	*p*-Value
BMI	In 1 Unit Change	1.5 (1.10–2.04)	0.0098
Allocation Group	Control	1.00	
	VESPA	0.05 (0.01–0.47)	0.0080

## Data Availability

The data are contained within the article.

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
