# Peer review of "Severity of Atelectasis during Bronchoscopy: Descriptions of a New Grading System (Atelectasis Severity Scoring System—“ASSESS”) and At-Risk-Lung Zones"

_diagnostics, 2024, doi:10.3390/diagnostics14020197_

Round 1
Reviewer 1 Report
Comments and Suggestions for Authors
Dear authors,
I read with great interest your manuscript regarding severity grading for atelectasis during bronchoscopy. Robotics and electromagnetic navigation, as well as the increasing number of peripheral lesions' identification, have made CT to body divergence especially due to atelectasis an upcoming issue that needs proper grading and management.
Abstract: Well written.
Introduction: Well written.
Methods: Well written.
Results: The results address the main question successfully. Discussion/Conclusion: The conclusions are consistent with the results. Overall comments: The article successfully addresses the issue of sufficient stratification of atelectasis during general anesthesia procedures. It is original, providing novel and important elements to interventional pulmonology field.Recommendation to be accepted in present form.
Best regards.
Author Response
We truly appreciate your reviewing our manuscript.
Reviewer 2 Report
Comments and Suggestions for Authors
The proposed paper completes the author's previous experience from the VESPA trial, adding new information. It is concise and well written. The main issue detected is the lack of sufficient references - I suggest adding, for example but not exclusivelly, papers presenting diagnostic yield for radial EBUS.
Author Response
Thank you for your review and suggestion. Since atelectasis can cause false positive RP-EBUS images, your point is quite relevant. We have added a sentence with regards to Tanner's Multicenter RCT on RP-EBUS with its corresponding citation.
Reviewer 3 Report
Comments and Suggestions for Authors
This is a study on the severity of atelectasis and the lung zones that were more commonly affected during bronchoscopy in the VESPA trial which multicenter randomized controlled trial. The authors demonstrated that 47% of patients had mild, 47% had moderate, and 6% had severe atelectasis. Higher BMI was associated with increased incidence of developing moderate to severe atelectasis. “ASSESS” is a simple method to categorize intra-bronchoscopy atelectasis which allows for a qualitative description of this phenomenon. Preventive strategies should be strongly considered in patients with risk factors for 30 atelectasis who have lesions located in zones 1 and 2, but not in zone 3. This study may provide some useful information on the severity and locations of atelectasis during bronchoscopy. I have some comments.
<Comments>
1. In line 35, Please describe the full term of the abbreviation, “CT”.
2. In line 55, Please describe the full term of the abbreviation, “FiO2”.
3. Please describe the study period.
4. It would be greatly appreciated if more information of the “VESPA trail” could be added briefly.
5. It would be better to record the results of this study using the tables in the result section. (for example, clinical characteristics, results of univariate and multivariable logistic regression)
Author Response
Thank you so much for your comments.
We described the abbreviations CT and FiO2, the study period (in Methods), and we added a bit more information on VESPA trial during the introduction.
There was not much more info on patient's characteristics (not that would be pertinent to the subject), but we did add 2 tables with Univariate and Multivariate analysis.
Thanks again.